



# Comparison of source apportionment approaches and analysis of
# non-linearity in a real case model application
Claudio A. Belis[1], Guido Pirovano[2], Maria Gabriella Villani[3], Giuseppe Calori[4], Nicola Pepe[4],
Jean Philippe Putaud[1]
[1] European Commission, Joint Research Centre, via Fermi 2748, 21027 Ispra (VA), Italy
[2] RSE Spa, via Rubattino 54, 20134, Milan, Italy
[3] ENEA Laboratory of Atmospheric Pollution, via Fermi 2748, 21027 Ispra (VA), Italy
[4] ARIANET s.r.l. via Gilino, 9 - 20128 Milan (MI) – Italy
*Correspondence to*: Claudio A. Belis (claudio.belis@ec.europa.eu)
**Abstract.** The response of particulate matter (PM) concentrations to emission reductions was analysed by assessing
the results obtained with two different source apportionment approaches. The brute force (BF) method source impacts,
computed at various emission reduction levels using two chemical transport models (CAMx and FARM), were
compared with the contributions obtained with the tagged species (TS) approach (CAMx with PSAT module). The
study focused on the main sources of secondary inorganic aerosol precursors in the Po Valley (Northern Italy):
agriculture, road transport, industry and residential combustion. The interaction terms between different sources
obtained from a factor decomposition analysis were used as indicators of non-linear $PM_{10}$ concentration responses to
individual source emission reductions. Moreover, such interaction terms were analysed in the light of the free ammonia
/ total nitrate gas ratio to determine the relationships between the chemical regime and the non-linearity at selected
sites. The impacts of the different sources were not proportional to the emission reductions and such non-linearity was
most relevant for 100% emission reduction levels compared with smaller reduction levels (50% and 20%). Such
differences between emission reduction levels were connected to the extent to which they modify the chemical regime
in the base case. Non-linearity was mainly associated with agriculture and the interaction of this source with road
transport and, to a lesser extent, with industry. Actually, the mass concentration of $PM_{10}$ allocated to agriculture by
TS and BF approaches were significantly different when a 100% emission reduction was applied. However, in many
situations the non-linearity in $PM_{10}$ annual average source allocation was negligible and the TS and the BF approaches
provided comparable results. PM mass concentrations attributed to the same sources by TS and BF were highly
comparable in terms of spatial patterns and quantification of the source allocation for industry, transport and residential
combustion. The conclusions obtained in this study for $PM_{10}$ are also applicable to $PM_{2.5}$.






**1. Introduction**
Air pollution is the main environmental cause of premature death. Ambient air pollution caused 4.2 million deaths
worldwide in 2016, contributing together with indoor pollution to 7.6% of all deaths (WHO, 2018). Air pollution
adverse health effects mainly occur as respiratory and cardiovascular diseases (WHO, 2016; EEA 2019). A key
element for the design of effective air quality control strategies is the knowledge of the role of different emission
sources in determining the ambient concentrations. This is usually referred to as source apportionment (SA) and
involves the quantification of the influence of different human activities (e.g. transport, domestic heating, industry,
agriculture) and geographical areas (e.g. local, urban, metropolitan areas, countries) to air pollution at a given location.
SA modelling studies involving secondary inorganic pollutants are generally based on chemistry transport models
(Mircea et al., 2020). Two different SA approaches are commonly used to allocate the mass of pollutants to the
different sources by means of chemical transport models:
• tagged species (TS) quantifies the contribution of emission sources to the concentration of one pollutant at
one given location by implementing algorithms to trace reactive tracers. SA studies based on tagging methods have
been carried out at both European scale (e.g. Karamchandani et al. 2017; Manders et al., 2017) and urban scale (e.g.
Pepe et al. 2019, Pültz, et al., 2019).
• brute force (BF or emission reduction impact) is a sensitivity analysis technique which estimates the change
in pollutant concentration (impact) that results from a change of one or more emission sources. Sensitivity analysis
techniques have been used to estimate the impact of different sources on pollution levels (e.g. Kiesewetter et al., 2015;
Thunis et al., 2016; Van Dingenen et al., 2018).
Even though these approaches are often considered as two alternative SA methods, they actually pursue different
objectives: TS aims to account for the mass transferred from the sources to the receptor in a specific area and time
window while BF is a sensitivity analysis technique used to estimate the response of the system to changes in
emissions. For a detailed discussion, refer to Belis et al. (2020); Mircea et al. (2020); Thunis et al. (2019).
Clappier et al. (2017) applied the concept of factor decomposition developed by Stein and Alpert (1993) to investigate
the differences between TS and BF using a theoretical example involving three sources. According to these authors,
the change in concentration of a given pollutant due to the change in the emissions of three sources A, B and C ($\Delta C_{ABC}$)
can be described as follows:
$$\Delta C_{ABC} = \Delta C_A + \Delta C_B + \Delta C_C + \hat{c}_{AB} + \hat{c}_{AC} + \hat{c}_{BC} + \hat{c}_{ABC} \qquad (1)$$
Where $\Delta C_A$, $\Delta C_B$ and $\Delta C_c$ are the variations of concentration of the studied pollutant due to the reduction of the single
sources A, B and C, respectively, and those coming from the interactions between these sources denoted by the terms
$\hat{c}_{AB}, \hat{c}_{AC}, \hat{c}_{BC}$ and $\hat{c}_{ABC}$ (see Appendix A for details). The interaction terms ($\hat{c}$) have the same units as the source impacts.
In the TS approach, the sum of the contributions of the various sources always matches the total pollutant concentration
by design. ($M_{poll} = M_A + M_B + M_C$), while this may be not the case for the BF approach ($\Delta C_{ABC} \neq \Delta C_A + \Delta C_B +$
$\Delta C_C$) under certain circumstances (Belis et al., 2020). The interaction terms in eq. 1 measure the consistency between
the sum of single emission sources with respect to the contemporary reduction of more than one source in BF, for
three sources $\Delta C_{ABC} - (\Delta C_A + \Delta C_B + \Delta C_C)$, which is an indicator of the non-linearity in the response of the pollutant
concentration to single source reductions (impacts).
There are different situations that may contribute to generating non-linear response when secondary pollutants'
precursors are emitted by different sources. They are double counting, chemical regime limited by one precursor,
competition between precursors, thermodynamic equilibrium between the secondary pollutant and its precursors, and
compensation. A detailed explanation of each of them is provided in Appendix A.
In the analysis of a theoretical example with three sources (agriculture, industry and residential), Clappier et al., 2017
observed that strong non-linearity is associated with secondary inorganic aerosol (SIA, ammonium nitrate and
ammonium sulfate) formation. However, this secondary aerosol may behave linearly or non-linearly depending on the





circumstances; for instance, the intensity of the emission reduction, which imposes the need to quantify it for different
emission reduction levels (ERLs) (see Section 3.2). Thunis et al. (2015) showed that for yearly average relationships
between emission and concentration changes, linearity is often a realistic assumption and consequently, TS and BF
methods are expected to provide comparable results, as reported by Belis et al. (2020). The abovementioned
considerations suggest the need to monitor whether non-linearity is significant for a given study area and time window.
The objective of this study is to identify and quantify the factors leading to non-linear response of PM concentrations
to source emission reductions in a real-world situation with significant PM concentrations. To that end, the influence
on $PM_{10}$ concentration of various sources with different chemical profiles were calculated using both the BF approach
with two different chemical transport models (CAMx and FARM) and the TS approach using one of these chemical-
transport models (CAMx).
The results of the simulations were then used to:
•   compare TS contributions with BF impacts
•   analyse the geographical patterns
•   compute interaction terms (of the Stein and Alpert algebraic expression) for the studied sources
•   compare the behaviour of various areas (urban, rural, etc.) with different chemical regimes
In this study, the focus is on the non-linearity associated with SIA formation, with particular reference to ammonium
nitrate ($NH_4NO_3$) and ammonium sulfate (($NH_4$)$_2SO_4$). The possible non-linear behaviour of any other PM component
(e.g. organics) is beyond the scope of this exercise.
**2.      Materials and methods**
The Po Valley was selected for this study because of its high levels of particulate matter due to the high emissions of
primary pollutants and precursors of SIA, whose high concentrations are also favoured by the stagnation of air masses
during the coldest months of the year (Belis et al., 2011, Larsen et al., 2012).
The air quality simulations were performed with CAMx (ENVIRON, 2016) and FARM (ARIANET, 2019) chemical
transport models (CTMs). Both are open-source modelling systems for multi-scale integrated assessment of gaseous
and particulate air pollution. Thanks to their variable spatial resolution they are used for urban to regional scale
applications, and simulating the atmospheric chemical reactions of the emitted precursors they allow reconstructing
the formation of most of the secondary compounds, including the constituents of particulate matter. CAMx is widely
used to assess the influence of pollution sources on air quality in a particular domain. The PM Source Apportionment
Technology (PSAT) ; Yarwood et al., 2004) implemented in CAMx offers the choice between several SA approaches,
which allows users  to easily compare e.g. TS vs BF methods for the estimation of source contributions to pollutant
concentrations using the same model. In addition, the application of the BF method with FARM made it possible to
evidence the structural behaviours that are less dependent on the specific model formulation and consequently to
obtain results of more general value.
The application of such CTMs required the implementation of a comprehensive modelling system (e.g. Pepe et al.,
2019), including specific tools aiming at creating the three main input categories: meteorological fields, emissions and
boundary conditions.
Both modelling systems were applied for the reference year 2010 over Northern Italy (Figures S1 and S2) considering
a computational domain that covers a 580 x 400 km$^2$ region, with a 5 km grid step. For the meteorological model WRF
(Skamarock et al., 2008) three nested grids were used, the largest one covering Europe and Northern Africa, and the
innermost one corresponding to Italy and Po Valley, respectively. The three meteorological domains have 45, 15, and
5 km grid resolution. For CTMs only the innermost WRF nested grid was used. Both CTMs were setup using the same
input meteorological data and horizontal grid structure of WRF. CTMs vertical grid was defined collapsing the 27





vertical layers used by WRF into 14 layers, while keeping identical the layers up to 1 km above ground level; in
particular, the first layer thickness was up to about 25 m from the ground like the corresponding WRF layer.
In CAMx, homogenous gas phase reactions of nitrogen compounds and organic species were reproduced through the
CB05 mechanism (Yarwood et al., 2005). The aerosol scheme was based on two static modes (coarse and fine).
Secondary inorganic compounds evolution were described by the thermodynamic model ISORROPIA (Nenes et al.,
1998), while SOAP (ENVIRON, 2011) was used to describe secondary organic aerosol formation. Meteorological
input data were provided by WRF and were completed by OMI satellite data (http://toms.gsfc.nasa.gov), including
ozone vertical content and aerosol turbidity. Vertical turbulence coefficients (Kv) were computed using O'Brien
scheme (O'Brien, 1970), but adopting two different minimum Kv values for rural and urban areas, so to consider heat
island phenomena and increased roughness of built areas.
FARM simulations were performed using the SAPRC-99 gas-phase chemical mechanism (Carter, 2000) and a three-
mode aerosol scheme (Binkowski and Roselle, 2003) including microphysics, ISORROPIA for thermodynamic
equilibrium of inorganic species and SORGAM (Schell et al., 2001) for secondary organic aerosol formation.
Meteorological input from WRF was complemented by Kv computed using Lange (1989) parameterisation.
Emissions were derived from inventory data at three different levels: European Monitoring and Evaluation Programme
data (EMEP, http://www.ceip.at/emission-data-webdab/emissions-used-in-emepmodels/) available over a regular grid
of 50 x 50 $km^2$ and ISPRA Italian national inventory data (http://www.sinanet.isprambiente.it/it/sia-
ispra/inventaria/disaggregazione-dellinventario-nazionale- 2010) which provides a disaggregation by province.
Moreover, regional inventories data based on INEMAR methodology (INEMAR – ARPA Lombardia, 2015) provided
detailed emissions data at municipality level for the four administrative regions of Lombardia, Piemonte, Veneto and
Emilia Romagna.
Each emission inventory was processed to obtain the hourly time pattern of the emissions. For the CAMx simulations
this was accomplished using the Sparse Matrix Operator for Kernel Emissions model (SMOKE v3.5) (UNC, 2013).
Temporal disaggregation was based on monthly, daily and hourly profiles deducted by CHIMERE (INERIS, 2006)
and EMEP models from Institute of Energy Economics and the Rational Use of Energy (IER) project named
GENEMIS (Pernigotti et al., 2013). Similar emission inventories processing was performed for FARM using Emission
Manager pre-processing system (ARIA Technologies and ARIANET, 2013).
Initial and boundary conditions were taken from a parent CAMx simulation covering the whole Italy and driven by
MACC-II system (http://www.gmes-atmosphere.eu/services/aqac/) that provides 3D global concentrations fields.
**Table 1: Macro-sectors according to EEA SNAP classification for emission inventories used to define air pollution sources**
**in this study**

| Source: SNAP Macrosector | SNAP Macrosector number | ABBREVIATION used in this study |
|---|---|---|
| Energy industry | 1 | OTHER |
| Residential and commercial/institutional combustion | 2 | RES |
| Industry (combustion & processes) | 3 and 4 | IND |
| Fugitive emissions from fuels | 5 | OTHER |
| Product use including solvents | 6 | OTHER |
| Road transport | 7 | TRA |
| Non-road transport | 8 | OTHER |
| Waste treatment | 9 | OTHER |
| Agriculture | 10 | AGR |


The CAMx modelling system was applied with the previously described setup in order to perform a TS run (with
PSAT) and three sets of BF runs with 100%, 50% and 20% emission reduction levels (ERLs) while FARM was used
to produce two sets of BF runs with 50% and 20% ERLs. Due to the high number of runs needed to apply the Stein



and Alpert decomposition only few sources were selected (Table 1). Originally, the study focused on the same system
of three sources (AGR, IND, RES) as the study by Clappier et al. (2017). However, due to the small non-linearity
associated with RES the focus was then shifted to a ternary system including AGR, TRA and IND. In total, 41 runs
were performed keeping all inputs as the base case, except for emissions that were modified according to the scheme
reported in Table 2.
In this study are mainly analysed the interactions between sources AGR, TRA and IND. Additional runs were executed
using FARM at 50% and 20% ERLs to test also the impacts and interactions of RES with the previous ones.
**Table 2: Sets of simulations performed in this study to compute the factor decomposition (Stein and Alpert, 1993). Every**
**set is named after the used CTM and ERL.**

| Simulation set<br>Reduced sources | CAMx 100% | CAMx 50% | CAMx 20% | FARM 50% | FARM20% |
|---|---|---|---|---|---|
| No reduction | Base case CAMx | | | Base case FARM | |
| AGR | x | x | x | x | x |
| IND | x | x | x | x | x |
| TRA | x | x | x | x | x |
| RES | | | | x | x |
| AGR-IND | x | x | x | x | x |
| AGR-TRA | x | x | x | | x |
| IND-TRA | x | x | x | x | x |
| RES-IND | | | | x | x |
| RES-TRA | | | | x | |
| RES-AGR | | | | x | |
| AGR-IND-TRA | x | x | x | | x |
| RES-IND-TRA | | | | x | x |

## 3.    Results and Discussion

### 3.1 Comparison between source apportionment TS and BF approaches

The yearly average $PM_{10}$ concentrations in the CAMx and FARM base case runs are shown in Figures S1 and S2 of
the supplementary material. Figure 1 shows the relative contributions of the modelled $PM_{10}$ sources using the TS
approach (CAMx-PSAT). The contributions of AGR are distributed across the entire Po Valley with maximum levels
in the centre and hot spots to the NW and SE. The IND contributions are the highest to the south, SE and NE of the
study area. The TRA contributions to $PM_{10}$ are the highest in the main urban areas, in particular Milan and Turin, and
along the main highways (e.g. A4 Turin - Venice). The highest contributions of all the other remaining sources
(OTHER) are observed in the Pre-alpine area and in the Alpine valleys (including some areas in the Apennines) where
the average $PM_{10}$ levels are lower than the Po Valley (Figures S1 and S2) and RES is an important source (see below).
The annual average impacts of AGR, TRA and IND on $PM_{10}$ derived by BF approach with CAMx and FARM for
different emission reduction levels (ERLs) are shown in Figure 2 while those of RES are shown in Figure S3. In a
linear situation the impacts allocated to each source decrease proportionally to the intensity of the emission reduction
($\Delta C_{100\%} = 2\ \Delta C_{50\%} = 5\ \Delta C_{20\%}$). For that reason, the impacts at the 100% ERL can be compared directly with TS
contributions while those of 50% and 20% must be multiplied by factor 2 and 5, respectively. The linearity between
different ERLs is discussed in Section 3.2. To facilitate the comparison between different models, impacts are
expressed as percentage of the base case in these figures. In Figure 2, the highest impacts are those of AGR followed
by TRA and IND. The output resulting from CAMx and FARM for the 50% and 20% ERLs present similar levels and
geographical patterns. Most of the highest impacts of AGR at 100% ERL are observed in or near the areas of high
$NH_3$ emissions (Figure S4), in which also TS points out high contributions of this source (Figure 1). However, in these
areas the BF impacts are nearly twice the TS contributions reported in Figure 1 (see also Figure 3, top left). Such high



levels could be attributed to a near double counting effect which is dominant only at this ERL because the effect of
limited chemical regime cannot be observed at 100% reduction (see Appendix A Section A2.2). At 50% and 20%
ERLs the impacts are lower than the 100% ERL, because of the limited regime, and the highest ones are located in
the mountainous areas (Alps and Apennines). Such pattern is likely due to the low emissions of the SIA precursors
($NH_3$, $NO_x$ and $SO_2$) (Figure S4) and the modest base case $PM_{10}$ concentrations in these areas. For IND and TRA, the
geographical patterns of BF are comparable to those of TS (Figure 1, Figure 3 left) and do not vary significantly
between the different ERLs, as discussed in Section 3.2. The only remark is that FARM presents higher TRA impacts
in the subalpine areas compared to CAMx, irrespective of the used SA approach.

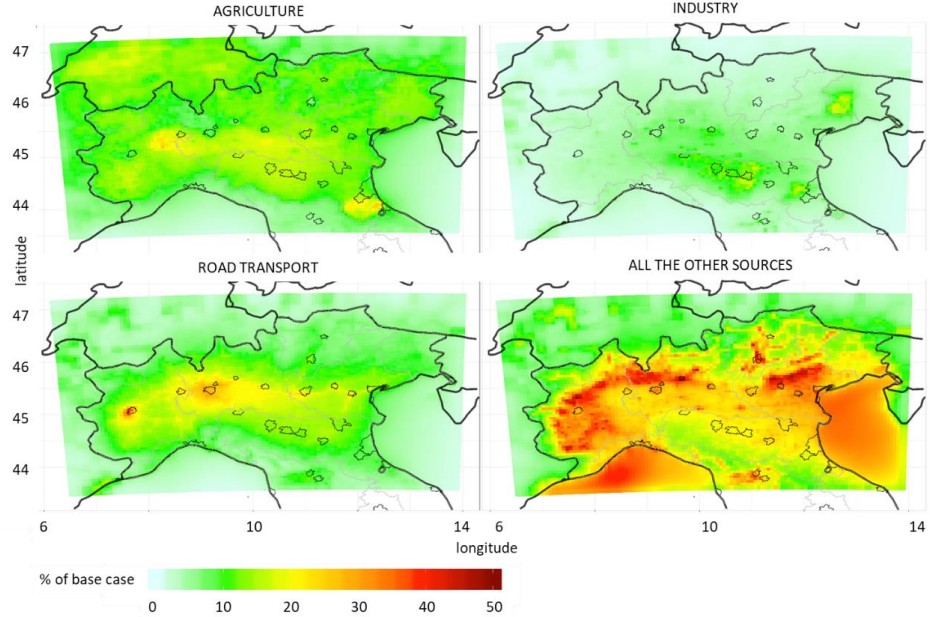


**Figure 1: Annual contributions of the $PM_{10}$ sources over the Po Valley area according to tagged species (TS) approach as
computed by CAMx PSAT. The grey lines indicate the boundaries of the regions and the polygons represent the municipal
areas of the main cities.**

As shown in Figure 3, the single grid cell annual average of BF impacts on $PM_{10}$ by IND and TRA plotted versus the
TS contributions are arranged on a line close to the identity indicating that BF and TS approaches lead to similar
results for these two sources. A similar behaviour is observed in all the ERLs even though the BF impacts estimated
with FARM present a higher dispersion than those obtained with CAMx. Such closer relationship between TS (CAMx-
PSAT) and CAMx BF results is likely a consequence of both being results of the same model. On the contrary, the
impacts of AGR on $PM_{10}$ at 100% ERL are more than twice the TS contributions in most grid cells, which is due to
the much greater AGR BF impacts on sulfate and nitrate than TS contributions at this ERL (Figures S5 and S6,
respectively). Such non-linear behaviour is associated with a situation near to double counting, which results in
negative interaction terms, and for nitrate, also to the $NH_4NO_3$ equilibrium, since both effects lead to BF impacts
higher than TS contributions (Appendix A).
Despite the comparable range of BF impacts and TS contributions of AGR on $PM_{10}$ at 50% and 20% ERLSs (Figure
3), there is a considerable dispersion around the regression line ($R^2$ between 0.65 and 0.72) indicating spatial
heterogeneity. In addition, impacts at 20% ERL present a slightly lower slope with respect to TS contributions than
those at 50% ERL. Also AGR BF impacts on nitrate present non-linear high values at 50% and 20% ERLs, which are
however compensated by ammonium impacts which are much lower than TS contributions (Figures S6 and S7,
respectively). The greater difference observed between TS and BF at 100% ERL for AGR compared to TRA and IND





are in part due to AGR being the only significant source of $NH_3$ in the domain. Consequently, a 100% reduction of
AGR implies an almost complete abatement of $NH_3$, while 100% reduction of TRA or IND does not reduce $NO_x$ and
$SO_2$ emissions completely (compensation effect). The reported differences between AGR TS contributions and BF
impacts on $PM_{10}$ concentrations are due to the way in which the two approaches allocate ammonium, nitrate and
sulfate to this source. TS allocates secondary constituents according to the mass of precursors deriving from each
source (Mircea et al., 2020; Yarwood et al., 2004). Therefore, for TS the contribution of AGR is close to the mass
fraction of ammonium in $PM_{10}$ and very little nitrate and sulfate is allocated to this source, since $SO_2$ and $NO_x$
emissions from AGR are small compared to those from IND and TRA. On the contrary, BF allocates these constituents
on the basis of the amount of $NH_4NO_3$ and/or $(NH_4)_2SO_4$ which is not formed when such sources are reduced.
Consequently, considerable nitrate and sulfate are allocated to AGR by BF, even though are not physically emitted by
this source, because there is no formation of $NH_4NO_3$ and/or $(NH_4)_2SO_4$ in the absence of $NH_3$ emissions from AGR.
Even in the cases where BF impacts and TS contributions to $PM_{10}$ are linear and close to identity, $PM_{10}$ constituents
may not behave in the same way. Sometimes, the linearity observed in $PM_{10}$ is the result of a compensation between
constituents for which BF impacts > TS contributions and others for which BF impacts < TS contributions. A good
example is TRA, whose annual BF impacts on $PM_{10}$ are aligned with TS contributions (Figure 3). However, the
ammonium impacts from this source are highly non-linear and larger than TS contributions (Figures S7), sulfate
impacts are quite non-linear and can be either larger or smaller compared to TS contributions (Figure S5), while nitrate
impacts are rather linear and slightly lower than TS contributions (Figure S6). A similar situation is observed for
nitrate and ammonium impacts from IND, with the difference that in this case sulfate, a component for which this
source is dominant, is rather linear.
The non-linearity between TS and BF source apportionment of $PM_{10}$ secondary inorganic constituents observed in
Figures S5 - S7 occur when the BF and TS approaches do not allocate these compounds to the same sources. For
instance, high non-linearity is observed for BF impacts of TRA and IND on ammonium because it is emitted almost
exclusively by AGR, while BF methods allocate impacts on ammonium to TRA and IND due to the atmospheric
reactions between $NH_3$ and $HNO_3$ or $H_2SO_4$, which are mainly emitted from TRA and IND, respectively. A similar
situation is observed for AGR impacts on sulfate and nitrate. TS allocates a negligible share of these compounds to
AGR (proportional to $SO_2$ and $NO_x$ emissions from AGR only), while the BF method allocates them to this source
proportionally to the $(NH_4)_2SO_4$ and $NH_4NO_3$ concentration variations, respectively.






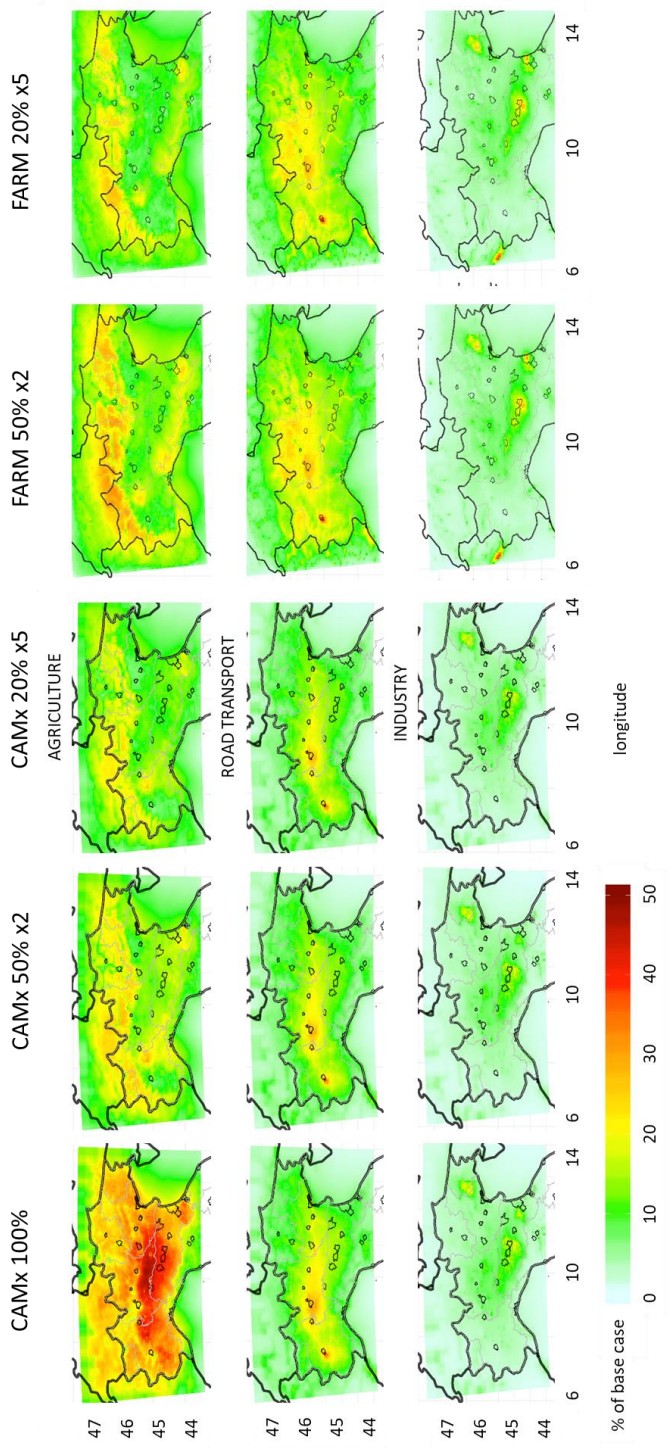


**Figure 2: Annual average impacts of AGR, TRA and IND expressed as proportion of the base case. From left to right CAMx 100%, 50% and 20% emission reduction**
**levels and FARM 50% and 20% emission reduction levels. For a direct comparison of the linearity between the different ERLs, the impacts of 50% and 20% are multiplied**
**by 2 and 5, respectively.**






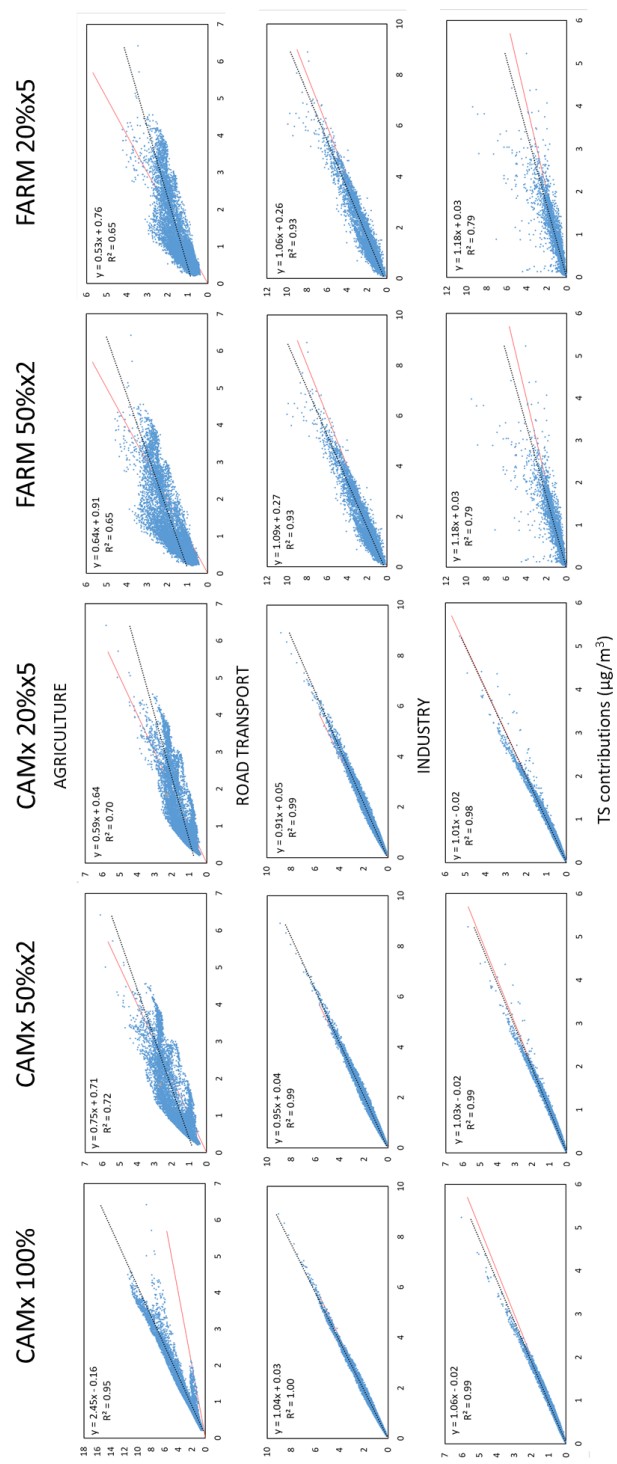

**Figure 3: Scatter plots of the single grid cell annual average BF source impacts (CAMx and FARM) on PM$_{10}$ versus the TS contributions (CAMx -PSAT) for 100%, 50% (multiplied by 2) and 20% (multiplied by 5) ERLs for AGR, TRA and IND. Dotted line: regression; red line: identity.**

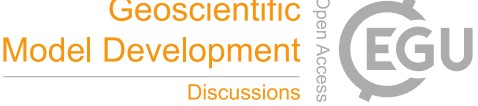




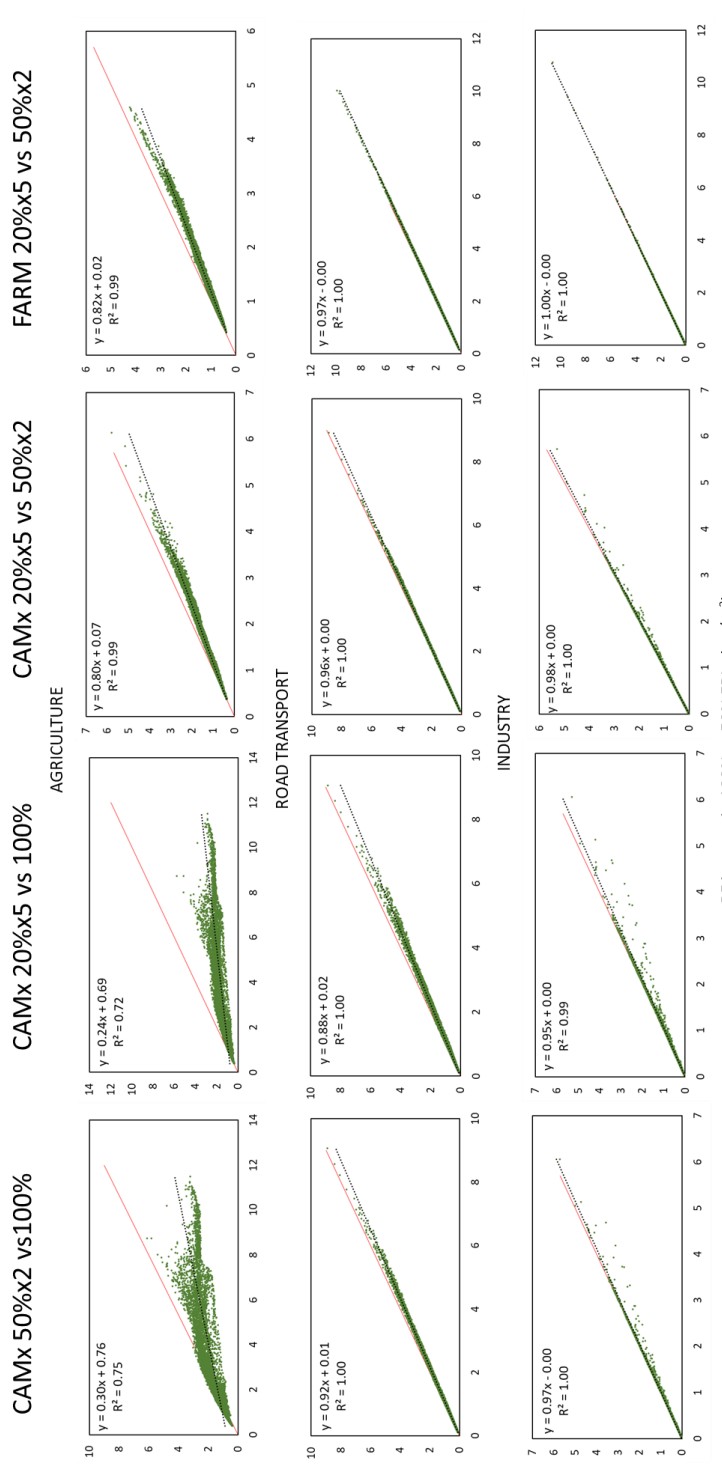

**Figure 4: Scatter plots of the single grid cell BF source impacts (CAMx and FARM) on PM$_{10}$ between the 100%, 50% (multiplied by 2) and 20% (multiplied by 5) ERLs for AGR, TRA and IND. Dotted line: regression; red line: identity**






The analysis of the impacts reported in this section clearly points out AGR as the source mostly associated with the
non-linear response of BF impacts with respect to TS.

## 3.2 Non linearity between different ERLs

In this section the connection between the magnitude of the emission reduction and the BF source impacts on $PM_{10}$ is
analysed more in detail. The scatter plots in Figure 4 depict the relationships between BF impacts at different ERLs
for every source and model. IND is the source for which the similarity between the different ERLs is the highest with
regression slopes and $R^2$ between impacts calculated for the three ERLs of CAMx and the two of FARM near unity.
Although also the regressions between TRA impacts are linear, the 50% ERL impacts are ca. 8% lower and the 20%
ERL ca. 12% lower than those obtained with 100% ERL using the same model. The impacts at 50% and 20% ERLs
are well correlated, and the latter are less than 5% below the former for both CAMx and FARM values. For AGR the
relationship between the impacts calculated for both 50% and 20% ERLs are clearly non-linear when compared to
100% ERL. In the latter impacts are 3 or 4 times higher than the former two, especially for mid to high impacts. By
comparison, the relationship between impacts at 50% and 20% ERLs is closer to linearity ($R^2 = 0.99$), with the latter
leading to 18% - 20% lower impacts than the former. The results shown in Figure 4 confirm that AGR is the source
presenting the most serious non-linearity among those emitting SIA precursors (see Section 3.1). In addition, the
analysis indicates that also for TRA the impacts of the different ERLs are not fully equivalent.
The large differences in AGR impacts on $PM_{10}$ between 100% and the other ERLs are likely explained by two reasons.
Firstly, turning off AGR 100% systematically shifts the system into a different chemical regime, while this is not the
case for the other sources, and secondly, the influence of limiting precursors (leading to less than double counting and
consequently less BF overestimation with respect to TS) is not expressed at 100% ERL (Appendix A Section A2.2).
The differences between 50% and 20% ERLs could be explained by the way in which limited chemical regimes
interact with the reduction of emissions. Since the non-linearity associated with limited chemical regimes appears only
when the emission reduction causes a drop of concentrations higher than the excess of the non-limiting precursor
(Appendix A), the chance of such non-linearity to influence source impacts is proportional to the emission reduction.
However, the relatively small differences observed between 50% and 20% ERLs are likely due to the smoothing effect
of the $NH_4NO_3$ equilibrium with respect to the non-linearity caused by a limited chemical regime because such
equilibrium leads $PM_{10}$ concentrations to change even when the non-limiting precursor emission reduction is lower
than the excess (Appendix A Figure A1).

## 3.3 Interaction terms

In Figure 5 are mapped the annual average interaction terms (ĉ) of the factor decomposition, which are used in this
study as indicators of the impact's non-linearity. The binary interaction terms are, in general, of higher magnitude
than the ternary interaction terms. The most negative interaction terms (indicating BF > TS) are observed in the 100%
ERL for the contemporary reduction of AGR and TRA in the rural areas located to the north of the Po Valley where
$NH_3$ is in excess, while the interaction terms are less negative in the main urban areas where $NH_3$ is a limiting factor.
When AGR and IND are both reduced 100%, the most negative interaction terms are observed in the industrial districts
around the main cities to the south of the Po Valley and to a lesser extent in the rural areas in the central Po Valley.
On the contrary, positive interaction terms are observed for the IND – TRA binary reduction due to the competition
between $HNO_3$ and $H_2SO_4$ that leads to an increase in the PM formation when $SO_2$ emissions (mainly industrial) are
reduced in presence of $NO_x$ (deriving mainly from road transport). Such maximum positive interactions are observed
in vast areas of the central Po Valley. A similar geographical pattern of the interaction terms is observed for 50% and
20% ERL (Figures S8 and S9, respectively) with the magnitude of the interaction decreasing with the emission
reduction.



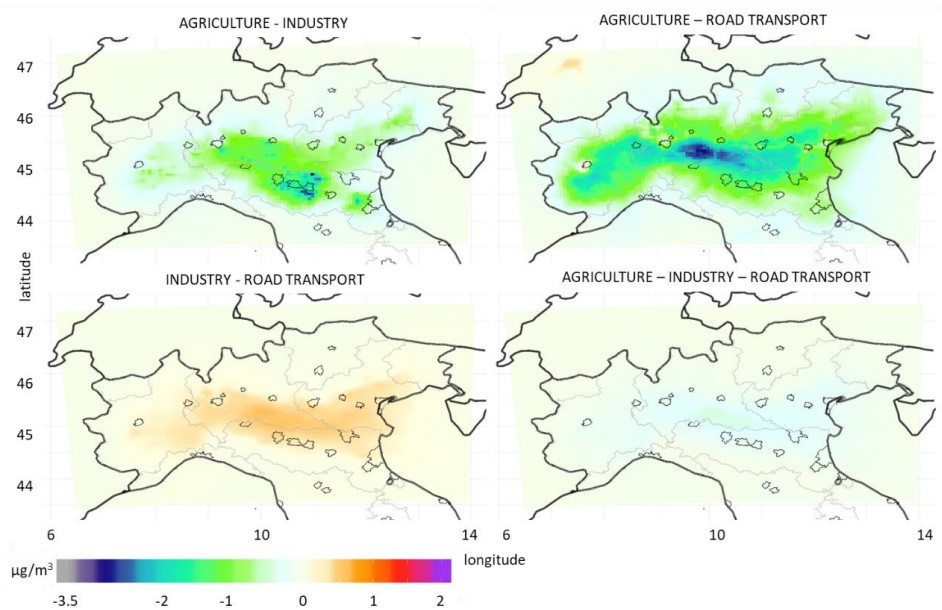


**Figure 5: Map of the binary and ternary interaction terms of the PM$_{10}$ factor decomposition for AGR, IND and TRA in the CAMx BF 100% scenarios.**

A similar analysis was carried with FARM at 50% ERLs for **residential heating** (Figure S10) and the resulting interaction terms were very low compared with those the other sources at the same ERL. The explanation is that despite the considerable contribution of this source to PM$_{10}$ its origin is mainly primary with a high non-reactive carbonaceous fraction (Piazzalunga et al., 2011) and therefore the impact on the secondary inorganic aerosol is limited.

The values of the interaction terms depend on the pollutant concentration. In order to define when ĉ is significantly different from zero, and consequently when the non-linearity is not negligible, the absolute value |0.5| % BC is proposed. Such arbitrary threshold was defined to highlight the interactions that according to the analysis of the impacts presented in the previous sections are associated with evident non-linear situations (e.g. AGR-TRA). In Figures S11 and S12 are reported the maps of the interaction terms expressed as % of the base case for 100% and 50% ERLs, respectively. According to the proposed threshold, at 100% ERL most of the Po Valley fall in the area where non-linearity is measurable for all the binary and ternary interactions. At 50% ERL, the non-linearity of the binary interactions AGR-IND are measurable in industrial districts located to the SW and NW of the Po Valley including the industrial areas to the NW of Milan. The non-linearity associated to the interaction AGR-TRA is not negligible in the entire Po Valley and also in the Alpine areas, probably due to the low PM$_{10}$ levels of the latter. The binary interaction IND-TRA exceeds the threshold only in the central area of the Po valley and in a hot spot to the NW of Milan. The ternary interaction is below the threshold for the entire domain. For the 20% ERL (not shown) all the interactions are negligible according to CAMx while FARM provides a pattern comparable to the 50% ERL.

### 3.4 Analysis of chemical regimes

A more in-depth analysis of the relationships between the chemical regime and the interaction terms was accomplished in three selected sites with different source emission set up (their position is shown in Figure S1). A rural location at the border between the provinces of Cremona and Brescia (CR_P) was selected because of the high NH$_3$ emissions while the local NO$_x$ and SO$_2$ emissions are very limited. The site of Milan (MI) was selected because representative of a typical urban situation with high NO$_x$ concentrations deriving from road transport emissions. The NH$_3$ emissions in this site are very limited and are associated with road transport while also SO$_2$ emissions are low and derive in part



325 from the energy production. The third site is an industrial area in the province of Ravenna (RA_P) located in the
326 South-Eastern Po Valley. In this location, there are considerable $SO_2$ emissions from industry, which also release $NO_x$,
327 and moderate $NH_3$ emissions from the agricultural sector. In order to define the chemical regime in each base case
328 (CAMx and FARM) and each of the simulations including binary or ternary interactions, the gas ratio (GR) proposed
329 by Ansari and Pandis (1998) was used:

330 $GR = ([NH_3] + [NH_4^+] - 2[SO_4^{2-}]) / ([HNO_3] + [NO_3^-])$    (3)

331 where concentrations are $nmol.m^{-3}$ or in nmol.mol of air (ppb).

332 The GR value defines three different chemical regimes:

333 (a) GR>1, in which $NH_4NO_3$ formation is limited by the availability of $HNO_3$,

334 (b) 0<GR<1, in which $NH_4NO_3$ formation is limited by the availability of $NH_3$, and

335 (c) GR<0, in which $NH_4NO_3$ formation is inhibited by $H_2SO_4$

336 The plots in Figure 6 display for each scenario the magnitude of the changes in the chemical regime with respect to
337 the base case, and the relationship between such changes and the interaction terms (expressed as a percentage of the
338 PM yearly mean concentrations). Each plot is divided in zones defined by the combination of the gas ratio (GR)
339 thresholds, and the threshold proposed in this study for the interaction terms (ĉ >|0.5% BC|) as indicator of non-
340 negligible non-linearity in the mass concentration allocated to sources with respect to the PM mass concentration.
341 A common feature of all three sites is that the higher the ERL the higher the difference between the GR of the scenarios
342 and the one of the base case providing evidence about the extent to which the emission reductions alter the original
343 conditions. The points representing simulations in which AGR is reduced sit to the left of their respective base case.
344 The scenarios with 100% ERL often lead to changes in the chemical regime and to the highest absolute interaction
345 terms. On the other hand, 50% and 20% ERLs lead, in general, to ĉ values closer to zero than 100% ERL, indicating
346 lower or negligible non-linearity (located in the white background area). All interactions IND-TRA give rise to ĉ
347 values ≥ 0, consistent with the competition effect (Appendix A Section A2.3). In CR_P and RA_P such simulations
348 lead to increase in GR (data points in Figure 6a and c are placed to the right of their base case), while in MI they lead
349 to null or slightly negative changes in GR (data points are located to the left of the base case in Figure 6b). This
350 behaviour indicates that the simultaneous reduction of IND and TRA leads to a higher impact of ammonia + nitric
351 acid on GR compared to the one of sulfate, in the three sites.

352 In CR_P the base cases of CAMx and FARM represent a $HNO_3$ limited chemical regime for $NH_4NO_3$ formation, in
353 line with the rural character of this area (Figure 6a). All scenarios where AGR is reduced lead to a decrease in GR
354 (points located to the left of the corresponding base case) indicating a loosening of the $HNO_3$ limitation, while all
355 those in which AGR is not reduced lead to an increase in GR (points located to the right of the corresponding base
356 case), indicating a stronger $HNO_3$ limitation. Sizeable negative ĉ are observed in scenarios reducing AGR 100%, likely
357 associated to the shift towards a $NH_3$ limited regime when AGR, the only significant source of this precursor, is turned
358 off. The described situation is reflected by the points representing the interaction terms AGR-IND (C10AI), AGR-
359 TRA (C10AT) and AGR-IND-TRA (C10AIT) of the 100% ERL located at the left-bottom of Figure 6a. The only
360 100% ERL scenario that does not lead to a chemical regime change is the contemporary reduction of IND and TRA
361 (C10IT). It also leads to positive interaction terms resulting from the competition between $HNO_3$ and $H_2SO_4$. In this
362 case, the abatement of $SO_2$ emissions leads to a reduced availability of $H_2SO_4$, which is replaced in the reaction with
363 $NH_3$ by $HNO_3$, the latter deriving from NOx emissions also from other sectors on top of TRA and IND (e.g. energy
364 industry) which is an example of compensation process (Appendix A Section A2.5). Figure 6a shows that for 50%
365 and 20% ERLs, the emission reductions do not modify the chemical regime at this site. The AGR-TRA (C5AT) is the
366 only scenario at 50% ERL leading to a non-negligible ĉ value. The scenarios at the 20% ERL generally show similar
367 behaviours as those at 50%.
368 In MI the base case simulations correspond to a chemical regime where $NH_4NO_3$ is limited by $NH_3$ (Figure 6b). The
369 inhibition of $NH_4NO_3$ formation by $H_2SO_4$ is unclear since the GR values calculated from both models are close to
370 the boundary between $H_2SO_4$ inhibited and non-inhibited chemical regimes. As in the previous site, all scenarios with
371 100% ERLs (C10) but one (C10IT) lead to a situation with strong $NH_3$ limitation, $H_2SO_4$ inhibition and negative

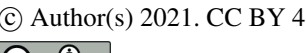



interaction terms (data points at the bottom left of Figure 6b). However, unlike the previous site, the combined 100%
reduction of IND and TRA (C10IT) in MI leads to a H$_2$SO$_4$ limited regime. Thus, all 100% ERL scenarios lead to a
strengthening of the H$_2$SO$_4$ inhibited chemical regime, which is relatively weak in the base case. As already observed
in CR_P, the interaction terms at 50% and 20% ERLs are negligible, with the exception of AGR – TRA (C5AT).
Among these scenarios, all those involving AGR reductions lead to regimes where NH$_4$NO$_3$ formation is limited by
NH$_3$ and inhibited by H$_2$SO$_4$ (data points to the left of the corresponding base case). On the contrary, most scenarios
not involving AGR (F5IT, F2IT, except C5IT) lead to situations where NH$_4$NO$_3$ formation is more limited by NH$_3$
(data points to the right of the corresponding base case) while the inhibition by H$_2$SO$_4$ is uncertain since data points
remain close to the boundary between the two regimes.

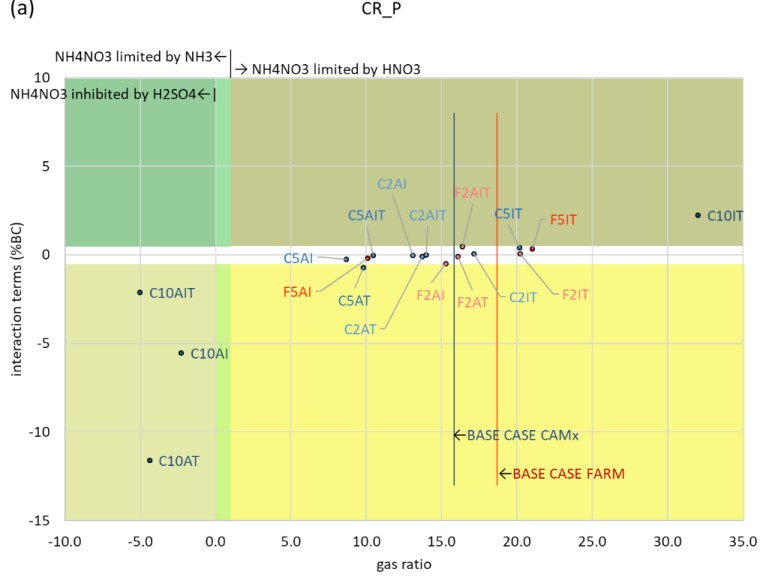






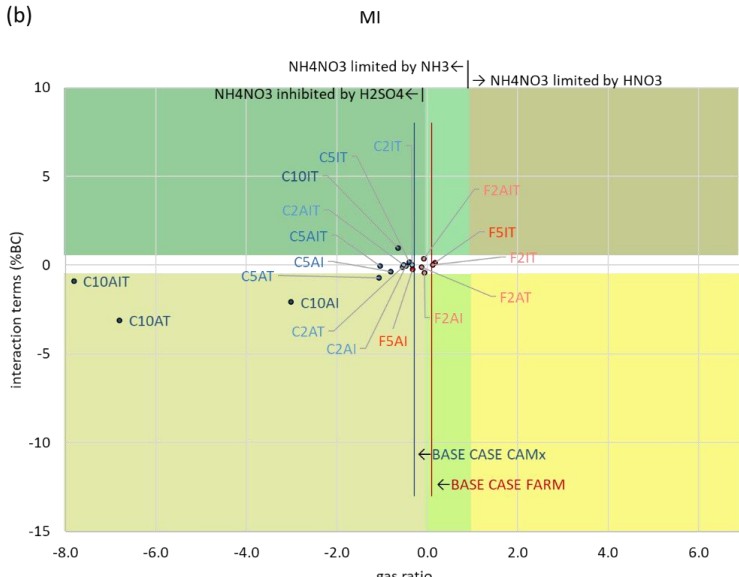


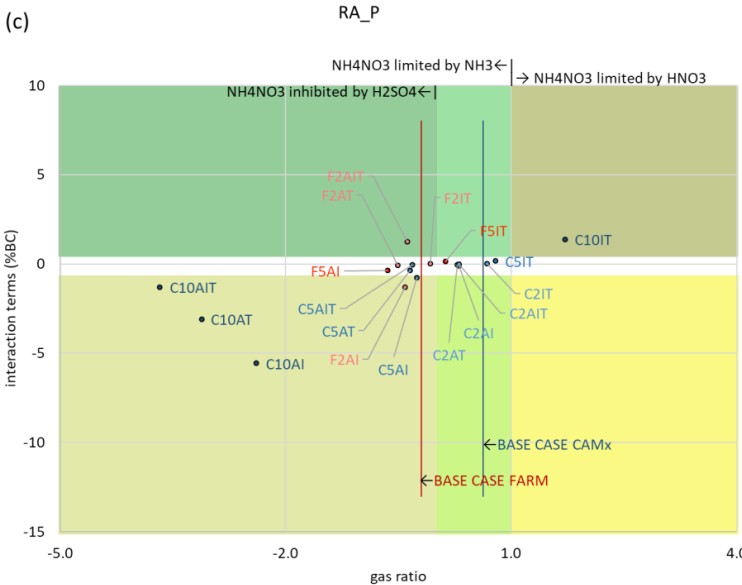


**Figure 6: Plot of the interaction terms (ĉ) in three selected sites with different chemical regimes versus the gas ratio (Ansari and Pandis, 1998). a) CR_P: Cremona province, b) MI: Milan and c) RA_P: Ravenna province. C: CAMx and F: FARM. 10, 5 and 2 indicate the 100%, 50% and 20% ERLs, respectively. A: agriculture, I: industry and T: transport. White background indicates negligible interaction terms.**

In RA_P, both base cases are in a regime of $NH_4NO_3$ formation limited by $NH_3$. However, for CAMx base case simulation $NH_4NO_3$ formation is not inhibited by $H_2SO_4$ while this is case for the FARM base case (Figure 6c). As in CR_P, the CAMx 100% scenarios in which AGR is reduced lead to decrease in GR and negative interaction terms



(data points at the bottom left), while the one involving the interaction IND-TRA (C10IT) leads to an increase in GR
and positive interaction term (data points at the top right). All scenarios in which AGR is reduced lead to $NH_3$
limitation and in most cases also $H_2SO_4$ inhibition chemical regimes (data points to the left of the respective base
case). On the contrary, the scenarios in which only combustion sources (TRA and IND) are reduced lead to regimes
where $NH_4NO_3$ formation is limited by $NH_3$ (data points to the right of the corresponding base case) and not inhibited
by $H_2SO_4$ (with some data points close to the boundary between the two regimes).
Among the scenarios at 50% and 20% ERLs, those involving AGR and IND lead to the highest absolute interaction
terms, of which some (C5AI, F2AI) are negative and clearly different from zero (non-linearity) with the exception of
F5AI that presents a negligible interaction term. The higher interaction terms for the AGR-IND scenarios with respect
to the other sites may be related to the greater importance of IND compared to TRA in this particular region.
**4.       Final remarks**
The theoretical analysis carried out by Clappier et al. (2017) applying factor decomposition was further developed in
this study by undertaking a real source apportionment exercise using a CTM model in an area with a complex
meteorology and chemistry, namely the Po Valley.
The **interaction terms** of the factor decomposition measure the consistency between the impacts obtained with single
source reductions compare to those of multiple source reductions. Consequently, they are also suitable indicators of
the non-linearity between the sum of the sources' mass concentration and the $PM_{10}$ total mass concentration. In
addition, the **interaction terms** used **in association with the GR** provide evidence about the relationships between
changes in the chemical regime (e.g. limiting precursor, competition) and the non-linear response of $PM_{10}$
concentrations to emissions reductions.
The analysis of the single secondary inorganic constituents of $PM_{10}$ combined with interaction terms and GR made it
possible to identify a series of mechanisms that influence the non-linear response of these pollutants when emission
reduction scenarios are applied to a real particulate pollution case: near double counting, precursor- limited chemical
regime, competition between precursors, thermodynamic equilibrium and compensation.
The results of this study confirm that due to the key role of $NH_3$ in the formation of SIA in the Po Valley, the **strongest**
**non-linear response of $PM_{10}$ concentrations to emissions reductions is associated with the AGR-TRA reduction**
**scenarios.** The differences in $PM_{10}$ attributed to AGR applying the TS and the BF approaches at 100% emission
reduction level **reach a factor 2**. On the other hand, the competition between $HNO_3$ and $H_2SO_4$ to react with $NH_3$
leads to a modest non-linear response of $PM_{10}$ in scenarios where TRA and IND are reduced simultaneously, especially
in areas with important $SO_2$ emissions. Tests carried out in the study area about RES indicate a very little non-linearity
associated with this source, likely due to the dominance of the primary fraction, including a considerable amount of
carbonaceous constituents.
The factors that trigger differences in SA between the TS and the BF approaches also lead to **non-linearity among**
**different levels of emission reduction**. For $PM_{10}$, this non-linearity is higher between 100% and the other reduction
levels and is mainly observed in scenarios involving AGR reductions where the differences may reach a factor of 3-
4, and to a lesser extent to scenarios involving TRA where differences are ca. 10%. This is due to a) the almost
complete suppression of $NH_3$ when turning off AGR while turning off TRA leaves other strong sources of $SO_2$ and
$NO_x$ active, and b) the fact that limiting precursors' effect is only observable for ERL below 100%. Moreover, the
present study shows that even when the **secondary inorganic components of $PM_{10}$ present a non-linear behaviour**
in their annual averages, the $PM_{10}$ response may result linear due to the compensation between different constituents.
It was also observed that in the majority of the tested scenarios at 50% and 20% ERLs, interaction terms are either
negligible or remain low (a few percent of the base case concentrations). In these conditions, the TS and the BF
approaches provide comparable results. Such findings were confirmed in this study by the direct comparison between



these two approaches that provided highly comparable spatial patterns and quantification of the role (contribution or
impact) of IND, TRA and RES sources.
The findings of the present work about $PM_{10}$ are also valid for the behaviour of **$PM_{2.5}$**. In the runs used for this study
these two size fractions present the same geographical patterns and values because the difference between them (the
coarse fraction) is mainly primary and thus expected to respond linearly to emissions reduction.

## 5. Conclusions

Considering the complexity of computing the Stein and Alpert decomposition for all the possible combinations of
source reductions (due to the high number of required runs), this work aims to provide a picture of the conditions that
give rise to non-linear response of $PM_{10}$ yearly averages for the reduction of single sources. Such picture is intended
as a contribution to simplify the tests needed in common modelling practice to detect non-linear responses by allowing
practitioners to focus on the situations that are more likely to be associated with non-linearity. Due to its high emission
levels and stagnation of air masses, the situations giving rise to non-linear responses are common in the Po Valley
making this region particularly suitable to study this kind of phenomena. The results of the study suggest that AGR is
the most important source from this point of view: a number of scenarios involving the reduction of emission from
AGR lead to non-linear responses of $PM_{10}$. This is due to the key role of $NH_3$, whose only significant source is AGR,
in the formation of secondary inorganic aerosol (SIA) in the test area. In addition, scenarios with high AGR emissions
reduction (e.g. 100%) lead to a shift of the $NH_4NO_3$ formation chemical regime. One of the implications of these
findings is that when there is a strong non–linear response (e.g. 100% reduction of AGR) it is not appropriate to sum
the impacts obtained with single source reductions to estimate the combined effect of more than one source.
Finally, it is important to stress the complementarity of the BF and TS techniques. TS is valuable when the focus is
on the actual mass transferred from sources to receptors in the situation described in the base case, while BF is the
best choice to test what is the response of the air quality system to changes in the emission rates.

## 6. Code and data availability

The model code and data used for the calculations and figures presented in this paper are available at
10.5281/zenodo.4306182.

## 7. Author contribution

C.A. Belis: conceptualisation, formal analysis, methodology, visualisation, writing – original draft preparation; G.
Pirovano: conceptualisation, formal analysis, review & editing; M.G. Villani:, formal analysis, visualization, review
& editing; G. Calori: formal analysis, visualization, review & editing; N. Pepe: formal analysis, visualization, review
& editing;  J.P. Putaud: conceptualisation, methodology, validation, review and editing

## 8. Competing interests

The authors declare that they have no conflict of interest

## 9. Acknowledgements

The authors acknowledge Kees Cuvelier for the development of a tool for the data elaboration and to Alain Clappier
and Philippe Thunis for the discussions during the preparatory phase of this work.



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





**Appendix A**
**A1) Interaction terms**
The interaction terms in the factor decomposition (Stein and Alpert, 1993) reflect the consistency between single
source emission reduction and contemporary reduction of more than one source and are indicators of the non-linear
response of particulate matter ($PM_{10}$ or $PM_{2.5}$) concentration to single source reductions.
**A1.1) Binary interactions**
Binary interactions describe the situation of two precursors $\alpha$ and $\beta$ emitted by two different sources A and B,
respectively, that react in atmosphere to form the secondary compound $\gamma$ ( $\alpha + \beta \rightarrow \gamma$). $\Delta C$ denotes the change in
the concentration of $\gamma$ as a consequence of applying the same percentage of reduction to sources A and B
separately or at the same time. The binary interaction term ($\hat{c}_{AB}$) is the difference between $\Delta C(\gamma)$ due the
contemporary reduction of both sources and the sum of $\Delta C(\gamma)$ due to the reduction of each single source:
$\hat{c}_{AB} = \Delta C_{AB} - \Delta C_A - \Delta C_B$                                              (A1)
**A1.2) Ternary interactions**
By analogy, ternary interactions refer to the interplay of three sources A, B and C each emitting one precursor
($\alpha$, $\beta$ and $\chi$, respectively) which react among each other in atmosphere for example as follows:
$\alpha + \beta \rightarrow \gamma_1$                                                                   (A2)
$2\alpha + \chi \rightarrow \gamma_2$                                                                   (A3)
$\gamma = \gamma_1 + \gamma_2$                                                                          (A4)
The ternary interaction term is a function of $\Delta C(\gamma)$ resulting from the reduction of all three sources at once, of
$\Delta C(\gamma)$ resulting from the reduction of each single source at a time, and of the $\hat{c}$ for all the combinations of binary
source reductions as described below (see also eq. 1):
$\hat{c}_{ABC} = \Delta C_{ABC} - \Delta C_A - \Delta C_B - \Delta C_C - \hat{c}_{AB} - \hat{c}_{AC} - \hat{c}_{BC}$                    (A5)
**A2) Situations giving rise to non-linearity**
This section analyses in detail the situations that may lead to non-linearity. Most of these situations are visible in
binary interactions, however, competition is only observable in ternary interactions. The different binary interactions
that are part of ternary interactions may represent different situations described in this section, some of which
leading to non-linearity and others not.
**A2.1) Double counting**
This interaction takes place when the concentrations of the emitted precursors ($\alpha$, $\beta$) are close to the stoichiometric
ratios and consequently none of them is limiting the reaction or is in excess. In addition, no compensation mechanisms
(see Section A2.5) take place and there are no other precursors competing for the reaction between $\alpha$ and $\beta$. Under
these circumstances, the application of the brute force (BF) approach leads to a 100% reduction of the concentration
of $\gamma$ when reducing the emissions of either source A or B by 100%. This is called "**double counting**" because the sum
of the scenario where only A is reduced by 100% and the one where only B is reduced by 100% is exactly the double
of the mass of the scenario when both sources A and B are reduced at once. This situation is described in the equation
below:





$\Delta C_{AB}$ = 1/2 ($\Delta C_A$ +$\Delta C_B$)                                                            (A6)
in other words, the $\Delta C$ of the contemporary reduction of A and B is the half of the sum of the $\Delta C$ of the single
reductions of A and B, respectively. In this situation, $\hat{c}_{AB}$ is negative and its absolute value is the highest and is equal
to the $\Delta C$ of A and B, which are equal to each other.
$\hat{c}_{AB}$ = -$\Delta C_A$ = -$\Delta C_B$ = -1/2 ($\Delta C_A$ +$\Delta C_B$)                              (A7)
A perfect double counting is a theoretical situation that does not take place in the "real-world" formation of secondary
inorganic aerosol (SIA) because of the influence of other factors such as reversible reactions and pH feedback on
solubility (deliquescent particles). Consequently, in this study we observe situations **near to double counting** where
the interaction terms are strongly negative, like the one described below.
Let's consider the reaction $NH_3 + HNO_3 \rightarrow NH_4NO_3$, where A is the source of $NH_3$ and B is the one of $HNO_3$ and
concentrations in ppb are denoted by $[NH_3] = a$ and $[NO_3] = b$. When setting Gas Ratio (GR, Ansari & Pandis, 1998)
$= 1$, $[SO_4^{2-}] = 0.5$ ppb (about 2 $\mu g.m^{-3}$) and assume particles to be deliquescent, then $d[PM]/d[NH_3] = 2.5$ and
$d[PM]/d[NO_3] = 0.6$. Under these circumstances, a 50% reduction of source A leads to a decrease in PM of $\Delta C_A = 2.5$
$\times a/2$; a 50% reduction of source B leads to a decrease in PM of $\Delta C_B = 0.6 \times b/2$; and a simultaneous 50% decrease of
emissions from both A and B leads to a PM decrease of $\Delta C_{AB} = a/2 + b/2$. The actual interaction term is:
$\hat{c}_{AB\_actual}$ = $\Delta C_{AB}$ - $\Delta C_A$ -$\Delta C_B$ = $- 0.75\ a + 0.2\ b$
while according to eq. (A7) the double counting interaction term is $\hat{c}_{AB\_DC}$ = -0.625 $a$ - 0.15 $b$
Since near the stoichiometric ratio $a$ is similar to $b$, the actual interaction term is close to but less negative than the
double counting interaction term.
**A2.2) Precursor limited chemical regime**
Most commonly, the concentrations of the precursors significantly differ from the stoichiometric ratio and
consequently one of them acts as limiting factor or limiting precursor (in the example below the one emitted by source
A, which implies $\Delta C_A > \Delta C_B$). In this case, the emission reduction can lead to two different situations:

2.2a) the reduction of the emissions causes a decrease of the non-limiting precursor ($\beta$) concentration lower or
equal to the its excess with respect to the limiting precursor ($\alpha$) leading to an interaction equal to zero because
$\Delta C_B$ is zero and $\Delta C_{AB} = \Delta C_A$.

$\hat{c}_{AB}$ = $\Delta C_{AB}$ - $\Delta C_A$ -$\Delta C_B$ = 0                                       (A8)

In this case **the potential interaction does not take place**

2.2b) the reduction of the emissions of source B is enough to reduce the concentration of precursor $\beta$ by more
than its excess with respect to $\alpha$ leading to a negative $\hat{c}_{AB}$ with lower absolute value than the double counting.

0 > $\hat{c}_{AB}$ > -1/2($\Delta C_A$ +$\Delta C_B$)                                                  (A9)

In this case there is a situation of **less than double counting**

Less than double counting is an intermediate situation between no interaction and the maximum interaction which is
the double counting and the interaction terms are always negative.
The limitation regime can only be observed when source reductions are less than 100% because, unless the same
precursor is emitted by other sources or transported from other areas (see Section A2.5), the complete removal of the
precursor leads to the complete removal of its products.



In the real world, situations where $NH_4NO_3$ formation is limited by free $NH_3$ availability (GR<1) or total nitrate
availability (GR>1) are common. However, due to feedback processes, the impact of reducing the emissions of a non-
limiting precursor is small but not null, while the one of reducing the emissions of a limiting precursor may be
smoothed by the $NH_4NO_3$ equilibrium (see Section A2.4).
**A2.3) Competition**
The interaction between two sources A and B can be affected by a third one C when the precursors emitted by the two
sources B and C compete to react with the one emitted by source A (See eqs. A2 and A3). In the formation of SIA,
there is competition between $HNO_3$ and $H_2SO_4$ to react with $NH_3$ to produce ammonium nitrate and ammonium
sulfate, respectively. $HNO_3$ derives from NOx emissions emitted i.a. by road transport (there are other sources), $H_2SO_4$
mainly comes from $SO_2$ emitted by industry, and $NH_3$ is mainly emitted from agriculture.
In situations where the formation of SIA is not limited neither by $H_2SO_4$ nor by $HNO_3$ availability (and conditions are
favourable to the formation of $(NH_4)_2SO_4$), the reaction $H_2SO_4 + NH_3$ produces 1 mol of $(NH_4)_2SO_4$ every 2 mols of
$NH_3$ while the reaction $HNO_3 + NH_3$ produces 1 mol of $NH_4NO_3$ for every mol of $NH_3$. The yield of aerosol in terms
of mols of the second reaction is twice the one of the first reaction.  The difference of mass in $\mu g/m^{-3}$ is as follows:
The reaction $2 NH_3 + H_2SO_4 \rightarrow (NH_4)_2SO_4$ leads to 3.9 $\mu g.m^{-3}$ PM from 1 $\mu g.m^{-3}$ $NH_3$.
The reaction $NH_3 + HNO_3 \rightarrow NH_4NO_3$ leads to 4.7 $\mu g.m^{-3}$ PM from 1 $\mu g.m^{-3}$ $NH_3$.
Consequently, when the $SO_2$ emissions are reduced in an $NH_3$-limited regime and $HNO_3$ replaces $H_2SO_4$ to react with
$NH_3$ there is an increase in the PM concentration.
In order to quantify the abovementioned competition it is necessary to compute the interaction between at least three
sources at once (eq. A5).
The competition in a three-source system may lead to negative $\Delta C$ (= increase in $PM_{10}$) for the single IND reduction
scenarios which results in positive binary IND-TRA interaction terms (see Section 3.4). The effect is also observed in
the TRA impact on sulfate and the IND impact on nitrate.
**A2.4) Equilibrium with solid $NH_4NO_3$**
The analysis of the previous cases is valid for unidirectional or irreversible chemical reactions. However, in the
atmosphere the reaction products, nitrate and ammonium, are in thermodynamic equilibrium with the reagents
ammonia and nitric acid:
$HNO_3 + NH_3 \leftrightarrow (NO_3^-, NH_4^+)$                                            (A10)
The actual concentrations of reagents and products depends on the ratio between the kinetics of the reaction in either
direction. For the conditions in which particulate ammonium nitrate is in solid state (non-deliquescent particles), the
equilibrium constant K of this reaction is the product of the reagent gas phase concentrations $[HNO_3(g)]$ and $[NH_3(g)]$:
$K = [HNO_3(g)] [NH_3(g)]$                                            (A11)
Any emission reduction leading to decreases in $HNO_3$ and/or $NH_3$ gas phase concentrations by a factor $q$ shall lead to
the shifting of the equilibrium towards the gas phase (volatilisation) of a concentration of ammonium nitrate $\Delta C$ so
that the equilibrium ($K = [HNO3(g)] \times [NH3(g)]$) is reached again.
If in the base case, the concentrations of the reagents are $a= [NH_3(g)]$ and $b= [HNO_3(g)]$ :
In case only the source of ammonia (A) is reduced, $\Delta C=\Delta C_A$ with $K= (b+\Delta C_A) (a/q+\Delta C_A)$                    (A12)
In case only the source of nitric acid precursors (B) is reduced, $\Delta C=\Delta C_B$ with $K= (b/q+\Delta C_B) (a +\Delta C_B)$                (A13)
In case both sources are reduced, $\Delta C=\Delta C_{AB}$ with $K= (a/q+\Delta C_{AB}) (b/q +\Delta C_{AB})$                    (A14)



Solving these second order equations for different emission reductions (represented by $q$ in eq. A 12-14) shows that
the inequality $\Delta C_{AB} < \Delta C_A + \Delta C_B$ (i.e. $\hat{c}_{AB} < 0$) is always observed (Figure A1). Moreover, the interaction terms vary
in a non-linear way with respect to the emission reduction becoming less negative when the system moves away from
stoichiometric conditions (Figure A1).

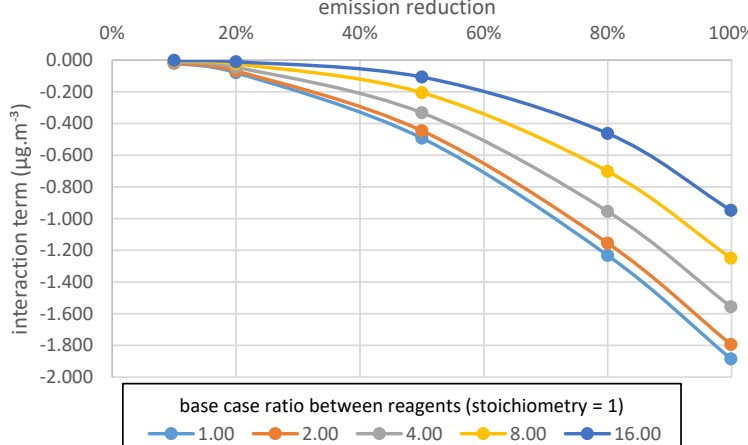


**Figure A1: Variation of the interaction terms as a function of the $NH_3$ and $HNO_3$ emissions reduction for different**
**stoichiometric ratios ranging from non-limited regime (r=1) to strongly limited regime (r=16). Calculations were performed**
**for conditions in which $K = 4$ ppb$^2$.**
**A2.5) Compensation**
In addition to the determinants described in the previous sections, which are mainly associated with the modellistic
approaches used to estimate source impacts and with atmospheric chemistry, there are other factors that may alter the
linearity of the relationship between the emission reductions $\Delta E$ and the response $\Delta C$. In this section, we generically
refer to such alterations as compensation.
Compensation are all the processes taking place in real world conditions which alter the $\Delta C$ expected to result from a
given $\Delta E$ in a theoretical exercise (either at the single cell or at the entire grid level), leading to interaction terms
different from those expected only on the basis of applied emission reduction.
**Compensation of precursor emissions:** the actual emission reduction ($\Delta E$) of one precursor is lower than the
expected $\Delta E$ in a system with few sources because in a complex system, like the one analysed in this study, there are
other sources of the same precursor in the grid. Consequently, the reduction of its concentration ($\Delta C$) may not be
proportional to the reduction ($\Delta E$) of one emission source.
**Compensation of precursor concentrations:** the actual $\Delta C$ is different from the one expected from $\Delta E$ because there
is import (advection) of this precursor from neighbouring grid cells or export (advection or deposition) from the
considered grid cell.
Below are presented examples on how the compensation may affect the interaction terms in different chemical
regimes.
a) The compensation alters the excess of the non-limiting precursor when emissions from not-considered sources or
advection from other cells contribute significantly to the concentration of this precursor and consequently prevent the
applied emission reduction from triggering a non-linear response (see Section A2.2).
b) The compensation alters the chemical regime. This can occur in different ways.





b1) Emissions from not-considered sources or advection processes are such that they keep the concentration of a
limiting precursor at the stoichiometric ratio with other precursors leading to  larger negative interactions terms than
those expected (see Section A2.1).
b2) Advection or deposition processes may reduce the level of a non-limiting precursor to levels close to the
stoichiometric ratio with other precursors and consequently lead to more negative interaction terms as described in
Section A2.1.
b3) Compensation may also alter the concentration of a precursor which is in competition with another. For instance,
when the emissions from three major sources (e.g. AGR, TRA, IND) are reduced, other sources (e.g. energy industry,
residential heating) may become predominant in controlling the chemical regime of SIA formation, which may result
in novel inhibition or competition situations (e.g. Section A2.4).