# Peer review of "Comparison of source apportionment approaches and analysis of"

_Geoscientific Model Development, 2020_

## Author Response (AR1)

Revision of manuscript gmd-2020-410

Answers to Reviewers

The authors would like to thank both reviewers for the time and effort they dedicated to check our manuscript and for their valuable remarks.

Comments from Referee #1 (Richard Kranenburg)

This paper gives a very good overview on the usage of two source apportionment methods (tagged species(TS) and Brute Force modelling (BF). Different usage and goals for both methods are well described, while synergy and understanding differences is essential to use the best of each method. I have two questions/suggestions which can lead to a braoder synergy between both methods and thus a better usage of both.

Author's response: We would like to thank R. Kranenburg for the useful comments and suggestions.

Section 3.1. In the TS approach, agriculture has a relative small contirbution to PM because it only accounts or the ammonium part, while traffic and industry accounts for the (heavier) nitrate and sulphate part. A redistribution of sources of NH4NO3 on molar basis will give a higher (and more fair, because NH3 is needed for formation) share to agriculture. This will also lead to a better agreement with BF scenarios because parts of the nitrate are now also accounted to agriculture. Please consider to report this issue in the discussion/final remarks.

Author's response: We agree with the reviewer that a TS approach, where sources are allocated on the basis of emitted mols rather than their mass, would smooth the effect of the different masses of the inorganic ions (ammonium, nitrate, sulphate). However, our impression is that it would not lead to the allocation of the ions to a different source. In other words, the TS approach will always associate agriculture only with ammonia because this is the only source emitting such pollutant.
In any case, testing this new SA method would require a completely new series of tests that go beyond the purpose of this paper.
Author's change in manuscript: We have now added the following comment in the Discussion (new lines 463-465):
"An option to emphasise the role of agriculture with this approach would be to develop a version based on the molar ratios instead of the mass. However, assessing the usefulness of such approach would require a new full set of tests."

Section 3.4 Please consider to add a map with gas-ratios. Especially, if it is posible to use this map to calculate a maximum ERL until which extent the chemical regime will stay the same in (almost) the full domain. With this calculation, the method is easy to implement on different areas and a indication can be given until which ERL the non-linearities are small and thus neglectible.

Author's response: We would like to thank the reviewer for this interesting suggestion that led us to make a more in-depth investigation. We agree that mapping an indicator of non-linearity based on chemical parameters would be very useful to predict the geographical areas in which the response is expected to be non-linear. In our study, the gas-ratio is used to demonstrate that there is a connection between the alteration of the chemical regime in one run compared to the base case and the non-linearity response in

that run. Therefore, the delta between the gas ratio in one run (scenario) and the base case could be an indicator of the non-linearity in that scenario (Δ gas-ratio). However, the examples presented in Figure 6 show that the relationship between the Δ gas-ratio magnitude and non-linearity varies from site to site. For instance, in CR_P (Figure 6a) non-linearity is associated with Δ gas-ratios of at list 20 units (approx.) while linear response corresponds to Δ gas-ratios of at most 8 units. On the contrary, in RA_P (Figure 6c) non-linearity appears in runs where the Δ gas-ratio is at least 2 units and the maximum Δ gas-ratio associated with linear response is less than 1 unit. In addition, there is one non-linear response run (F2AI) presenting the same levels of Δ gas-ratios observed in linear response runs. These examples show that a map of the Δ gas-ratio would be very difficult to read because the quantitative relationship between this indicator and the non-linear response varies from site to site.

In any case, we would like to recall that the maps of the interaction terms (Figure 5) summarise quite well the non-linear response in the different areas of the domain.

Author's change in manuscript: We have now added the following paragraph in section 3 (new lines 401-403):

"The numerical relationship between the interaction terms and the gas ratio delta (i.e. the difference between the gas ratio in one run and the corresponding base case) varies from site to site and, therefore, it is not possible to define acceptability thresholds valid for the entire domain."

Further I will mention a few technical corrections

line 158: --> In tis study the interactions between .. are analysed

Author's response: thank you for pointing this out
Author's change in manuscript: the sentence was modified as suggested.

line 285: In Figure 5 the annual interaction … are mapped

Author's response: thank you for suggesting this improvement
Author's change in manuscript: the sentence was modified as suggested.

Section A1.1 and A1.2 are intended, while A2.1 and A2.2 are not

Author's response: thank you for pointing out this typo
Author's change in manuscript: the indentation of sections A1.1 and A1.2 was corrected.

Reviewer 2

In this manuscript, the authors identify and quantify the factors leading to a non-linear response of PM concentrations to source emission reductions in a real-world situation. The paper is very well structured, it includes an extensive and comprehensive analysis of results, well presented and discussed, despite the complexity of the subject. This type of study and the results presented here are of high scientific interest. It is of great importance for the research modelling community. Please find below some minor comments and suggestions to improve this manuscript

In the introduction, the authors state that knowledge of the role of different sources of emissions in determining environmental concentrations is a key element in the design of effective air quality control strategies. However, throughout the manuscript, it is not explained how. It would be interesting to address this relationship in the conclusions section, demonstrating that this work is not only a purely scientific/academic investigation, but that it is of great interest to the general society.

Author's response: We are grateful to the reviewer for the appreciation of the manuscript and the constructive remarks. We agree with the reviewer about the need to better link the technical findings of this study with the practical application and it usefulness for the general society.

Author's change in manuscript: we have restructured the Conclusions and added the following new paragraphs explaining the practical implications of the study findings and their application for the design of more effective pollution abatement strategies (new lines 458-483).

"BF and TS are different but complementary techniques. Understanding how they work is necessary to adopt the one which is most suitable for the purposes of the work. On the one hand, BF is the best choice to assess the response of the air quality system to changes in the emission rates. For instance, this approach emphasises better the key role of agriculture and is then most suitable for planning purposes. On the other hand, TS is most valuable when the focus is on the actual mass transferred from sources to receptors in the situation described in the base case. It is, therefore, most appropriate for studying the health impact of sources because the effect of pollutants depends on the dose. An option to emphasise the role of agriculture with this approach would be to develop a version based on the molar ratios instead of the mass. However, assessing the usefulness of such approach would require a new full set of tests.

One of the main outcomes of this study is that in most situations (linear response) the two approaches provide similar results for the annual averages, which is the time averaging required for long-term air quality indicators. However, for shorter time windows (daily, seasonal averages or pollution episodes) the non-linearities are likely be more prominent. If there is a clear non-linear response, precaution is needed in the interpretation of the results from both approaches:

- in BF it is not appropriate to sum of the impact of the sources obtained by single source reduction because they may not match the total PM while

- in TS there could be a distortion in the allocation of secondary aerosol because it does not account for indirect effects (Mircea et al, 2020; Thunis et al., 2019).

Moreover, in case of non-linear responses, also extending the results of BF for a specific ERL to another (e. g. 20 to 50 or 100%) could be misleading.

To overcome the limitations of strong non-linear responses on source apportionment the only option is to run a scenario analysis with the exact combination of emission reductions for all the sources at once so all the interactions among them leading to secondary compounds are accounted for. However, this approach is valid only for one specific situation.

The methodology proposed in this study provides the means to identify non-linear responses to promote a more mindful use of source apportionment techniques. The ultimate goal of which is to inform more

effective air quality plans with a consequent more efficient use of economic resources and a faster achievement of air quality standards to protect human health and ecosystems."

Why did the authors choose to include a Final Remarks section and a Conclusion section? Both contents are suitable as final remarks or as conclusions. I would suggest to merge the two sections, or include the final remarks as a last part of the discussion.

Author's response: We thank the reviewer for this suggestion

Author's change in manuscript: We have now merged the two sections under the title "Conclusions".

Line 306: Authors mention BC, although BC acronym is not defined in the text. It stand for Black Carbon right? Please clarify and add it to the text.

Author's response: BC means base case. We apologise for the confusion.

Author's change in manuscript: We have now defined the acronym at the first appearance of the term (new line 156).

Figures S1, S4, S11, S12 (Supplementary Material): The colour scales used in the figures are too small and difficult to read.

Author's response: We agree with this remark

Author's change in manuscript: We have enlarged the legend of the colour scales in figures S1, S4, S11 and S12.

Additionally, in figures S11 and S12 it would be interesting to maintain the same colour scale in the different maps for each figure so that the differences were more evident and easier to identify

Author's response: When we use the same colour scale for all the maps in the same figure, the gradients are not visible any longer in those with the smaller value range (you just see a uniform background in the whole map). Therefore, we concluded that the figure is more clear if we keep different colour scales for each map.

Author's change in manuscript: We have enlarged the entire figures S11 and S12 to improve the visualisation of the colour gradients in the maps.